# SECURE FEDERATED LEARNING OF USER VERIFICATION MODELS

## ABSTRACT

We consider the problem of training User Verification (UV) models in federated setup, where the conventional loss functions are not applicable due to the constraints that each user has access to the data of only one class and user embeddings cannot be shared with the server or other users. To address this problem, we propose Federated User Verification (`FedUV`), a framework for private and secure training of UV models. In `FedUV`, users jointly learn a set of vectors and maximize the correlation of their instance embeddings with a secret user-defined linear combination of those vectors. We show that choosing the linear combinations from the codewords of an error-correcting code allows users to collaboratively train the model without revealing their embedding vectors. We present the experimental results for user verification with voice, face, and handwriting data and show that `FedUV` is on par with existing approaches, while not sharing the embeddings with other users or the server.

## 1 INTRODUCTION

There has been a recent increase in the research and development of User Verification (UV) models with various modalities such as voice (Snyder et al., 2017; Yun et al., 2019), face (Wang et al., 2018), fingerprint (Cao & Jain, 2018), or iris (Nguyen et al., 2017). Machine learning-based UV features have been adopted by commercial smart devices such as mobile phones, AI speakers and automotive infotainment systems for a variety of applications such as unlocking the system or providing user-specific services, e.g., music recommendation, schedule notification, or other configuration adjustments (Matei, 2017; Barclays, 2013; Mercedes, 2020).

User verification is a binary decision problem of accepting or rejecting a test example based on its similarity to the user's training examples. We consider embedding-based classifiers, in which a test example is accepted if its embedding is close enough to a reference embedding, and otherwise rejected. Such classifiers are usually trained with a loss function that is composed of two terms, 1) a positive loss that minimizes the distance of the instance embedding to the positive class embedding, and 2) a negative loss that maximizes the distance to the negative class embeddings. The negative loss term is needed to prevent the class embeddings from collapsing into a single point (Bojanowski & Joulin, 2017).

Verification models need to be trained with a large variety of users' data so that the model learns different data characteristics and can reliably reject imposters. However, due to the privacy-sensitive nature of the biometric data used for verification, it is not possible to centrally collect large training datasets. One approach to address the data collection problem is to train the model in the federated setup, which is a framework for training models by repeatedly communicating the model weights and gradients between a central server and a group of users (McMahan et al., 2017a). Federated learning (FL) allows training models without users having to share their data with the server or other users and, hence, helps enable private training of verification models.

Training UV models in federated setup, however, imposes additional constraints that each user has access to the data of only one class, and cannot share their embedding vector with the server or other users due to security reasons. Specifically, sharing embeddings with others can lead to both training- and test-time attacks. For example, the server or other users can run a poisoning attack (Biggio et al., 2012) and train the model so that it verifies fake examples as the examples generated by a particular

user. They can also carry out evasion attacks at test time (Biggio et al., 2013; Szegedy et al., 2013) to fool the model to verify impostors as a target user.

Without having access to embedding vectors of others, users cannot compute the negative loss term for training the model in the federated setup. A recent work (Yu et al., 2020) studied the problem of federated learning with only positive labels and proposed `FedAwS`, a method that allows users and the server to jointly train the model. In `FedAwS`, at each round, users train the model with the positive loss function and send the new models to the server. The server then updates the averaged model using an approximated negative loss function that maximizes the pairwise distances between user embeddings. `FedAwS` keeps the embedding of each user private from other users but reveals all embeddings to the server, which, as mentioned earlier, undermines the security of verification models in real-world applications.

In this paper, we propose Federated User Verification (`FedUV`), a framework for training UV models in federated setup using only the positive loss term. Our contributions are summarized in the following.

- We propose a method where users jointly learn a set of vectors, but each user maximizes the correlation of their instance embeddings with a secret linear combination of those vectors. We show that, under a condition that the secret vectors are designed with guaranteed minimum pairwise correlations, the model can be trained using only the positive loss term. Our framework, hence, addresses the problem of existing approaches where embeddings are shared with other users or the server (Yu et al., 2020).

- We propose to use error-correcting codes to generate binary secret vectors. In our method, the server distributes unique IDs to the users, which they then use to construct unique vectors without revealing the selected vector to the server or other users.

- We present a verification method, where a test example is accepted if the correlation of the predicted embedding with the secret vector is more than a threshold, and otherwise rejected. We develop a "warm-up phase" to determine the threshold for each user independently, in which a set of inputs is collected and then the threshold is computed so as to obtain a desired True Positive Rate (TPR).

- We present the experimental results for voice, face and handwriting recognition using Vox-Celeb (Nagrani et al., 2017), CelebA (Liu et al., 2015) and MNIST-UV datasets, respectively, where MNIST-UV is a dataset we created from images of the EMNIST dataset (Cohen et al., 2017). Our experimental results show that `FedUV` performs on par with `FedAwS`, while not sharing the embedding vectors with the server.

## 2 BACKGROUND

### 2.1 FEDERATED LEARNING

Consider a setting where $K$ users want to train a model on their data. Federated learning (FL) allows users to train the model by the help of a central coordinator, called server, and without sharing their local data with other users (or the server). The most commonly-used algorithm for FL is Federated Averaging (`FedAvg`) described in Algorithm (1) (McMahan et al., 2017a).

### 2.2 USER VERIFICATION WITH MACHINE LEARNING

User verification (UV) is a binary decision problem where a test example is accepted (reference user) or rejected (impostor user) based on its similarity to the training data. We consider embedding-based classifiers,

---

**Algorithm 1** (McMahan et al., 2017a) `FedAvg`.
$\theta_t$: model parameters at round $t$, $K$: number of users, $\epsilon$: fraction of users selected at each round, $D_u$: dataset of user $u$ with $n_u$ examples.

**FedAvg:**
    **Server:** Initialize $\theta_0$
    **Server:** $\kappa \leftarrow \max(\epsilon \cdot K, 1)$
    **for** each global round $t = 1, 2, \ldots$ **do**
        **Server:** $S_t \leftarrow$ (random set of $\kappa$ users)
        **Server:** Send $\theta_{t-1}$ to users $u \in S_t$
        **Users** $u \in S_t$: $\theta_t^u, n_u \leftarrow$ UserUpdate$(\theta_{t-1}, D_u)$
        **Server:** $\theta_t \leftarrow \frac{\sum_{u \in S_t} n_u \theta_t^u}{\sum_{u \in S_t} n_u}$

**UserUpdate**$(\theta, D)$**:** *// Done by users*
    $\mathcal{B} \leftarrow$ (split $D$ into batches of size $B$)
    **for** each local epoch $i$ from 1 to $E$ **do**
        **for** batch $b \in \mathcal{B}$ **do**
            $\theta \leftarrow \theta - \eta \nabla \ell(\theta; b)$
    return $\theta$ and $|D|$ to server

---

in which both the classes and the inputs are mapped into an embedding space such that the embedding of each input is closest to the embedding of their corresponding class. Let $w_y \in \mathbb{R}^{n_d}$ be the embedding vector of class $y$ and $g_\theta : \mathcal{X} \to \mathbb{R}^{n_d}$ be a network that maps an input $x$ from the input space $\mathcal{X}$ to an $n_d$-dimensional embedding $g_\theta(x)$. Let $d$ be a distance function. The model is trained on $(x, y)$ so as to have $y = \arg\min_u d(g_\theta(x), w_u)$ or, equivalently,

$$d(g_\theta(x), w_y) < \min_{u \neq y} d(g_\theta(x), w_u). \tag{1}$$

Hence, the loss function can be defined as follows:

$$\ell(x, y; \theta, w) = d(g_\theta(x), w_y) - \lambda \min_{u \neq y} d(g_\theta(x), w_u). \tag{2}$$

Minimizing the loss function reduces to minimizing the distance of the instance embedding to the true class embedding and maximizing the distance to the embeddings of other classes. The two terms are called positive and negative loss terms, respectively. The negative loss term is needed to ensure that the training does not lead to a trivial solution that all inputs and classes collapse to a single point in the embedding space (Bojanowski & Joulin, 2017).

### 2.3 Error-Correcting Codes

Error correcting codes (ECCs) are techniques that enable restoring sequences from noise. A binary block code is an injective function $C : \{0, 1\}^m \to \{0, 1\}^c, c \geq m$, that takes a binary message vector and generates the corresponding *codeword* by adding a structured redundancy, which can be used to obtain the original message from the corrupted codeword. In ECCs, each message bit affects many transmitted symbols in such a way that the corruption of some symbols by noise usually allows the original message bits to be extracted from the other, uncorrupted received symbols that also depend on the same message bits.

ECCs are designed to maximize the minimum Hamming distance, $d_{\min}$, between distinct codewords, where Hamming distance between two sequences is defined as the number of positions at which they differ. A code with minimum distance $\delta$ allows correcting up to $(\delta - 1)/2$ errors (Richardson & Urbanke, 2008). Also, with the same code length, $c$, the minimum distance decreases as the message length, $m$, increases. In this paper, we use binary BCH codes which are a class of block codes with codewords of length $c = 2^i - 1, i \geq 3$ (Bose & Ray-Chaudhuri, 1960). Note that the choice of the coding algorithm is not crucial to our work and our method works with any ECC algorithm.

## 3 User verification with Federated Learning

In this section, we outline the privacy and security requirements of the training UV models and describe the challenges of training UV models in the federated setup.

### 3.1 Privacy and Security Requirements of Training UV Models

Verification models need to be trained with a large variety of users' data so that the model learns different data characteristics and can reliably verify users. For example, speaker recognition models need to be trained with the speech data of users with different ages, genders, accents, etc., to be able to reject impostors with high accuracy. One approach for training UV models is to collect the users' data and train the model centrally. This approach is, however, not privacy-preserving due to the need to have direct access to the users' biometric data.

An alternative approach is using FL framework, which enables training with the data of a large number of users while keeping their data private by design. Training UV models in federated setup, however, poses its own challenges. As stated in Section (2.2), training embedding-based classifiers requires having access to all class embeddings to compute the loss function in (2). In UV applications, however, class embeddings are used for the verification and, hence, are considered security-sensitive information and cannot be shared with the server or other users.

Providing security is particularly important in UV applications, where the model might be trained and deployed in adversarial settings. Specifically, the leakage of the embedding vector makes the verification model vulnerable to both training- and test-time attacks, examples of which are provided in the following.

- Poisoning attack (Biggio et al., 2012): The server participates in training and trains the model with the loss function $d(g_\theta(x_u^*), w_u)$ for some $x_u^*$. At test time, the model generates $w_u$ for the input $x_u^*$ and thus wrongly verifies $x_u^*$ as a true example from user $u$.
- Evasion attack (Biggio et al., 2013; Szegedy et al., 2013): Attacks based on adversarial examples are known to be highly effective against deep neural networks (Carlini & Wagner, 2017). In the context of UV models, when a target embedding vector is known, an evasion attack can be performed to slightly perturb any given input such that the predicted embedding matches the target embedding and thus is accepted by the model.

## 3.2 PROBLEM STATEMENT

Without the knowledge of the embedding vectors of other users, users cannot compute the negative loss term in (2) for training the model in federated setup. Training only with the positive loss function also causes all class embeddings to collapse into a single point. In this paper, we address the following questions: 1) how to train embedding-based classifiers without the negative loss term? and 2) how this can be done in the federated setup?

## 3.3 RELATED WORK: FEDERATED AVERAGING WITH SPREADOUT (FedAwS)

In embedding-based classifiers, the negative loss term maximizes the distance of instance embeddings to the embeddings of other classes. A recent paper (Yu et al., 2020) observed that, alternatively, the model could be trained to maximize the pairwise distances of class embeddings. They proposed Federated Averaging with Spreadout (FedAwS) framework, where the server, in addition to averaging the gradients, performs an optimization step to ensure that embeddings are separated from each other by at least a margin of $\nu$. Formally, in each round of training, the server applies the following geometric regularization $\text{reg}_{\text{sp}}(W) = \sum_{u \in [K]} \sum_{u' \neq u} (\max(0, \nu - d(w_u, w_{u'})))^2$. FedAwS eliminates the need for users to share their embedding vector with other users but still requires sharing it with the server, which seriously undermines the security of the real-world verification models.

## 4 PROPOSED METHOD

### 4.1 TRAINING WITH ONLY POSITIVE LOSS

Training UV models using the loss function in (2) requires users to jointly learn the class embeddings, which causes the security problem of sharing the embeddings with other users. To address this problem, we propose a method where users jointly learn a set of vectors, but each user maximizes the correlation of their instance embedding with a secret linear combination of those vectors. The same linear combination is also used for user verification at test time.

Let $W \in \mathbb{R}^{c \times n_d}$ be a set of $c$ vectors and $v_u \in \{-1, 1\}^c$ be the secret vector of user $u$. We modify the loss function in (2) as follows:

$$\ell(x, y, v; \theta, W) = \ell_{\text{pos}} + \lambda \ell_{\text{neg}}, \tag{3}$$
$$\text{where } \ell_{\text{pos}} = d(g_\theta(x), W^T v_y) \text{ and } \ell_{\text{neg}} = -\min_{u \notin y} d(g_\theta(x), W^T v_u).$$

Let us call $s_u = W^T v_u$ the *secret embedding* of user $u$. Note that users still need to know the secret vector, $v_u$, or the secret embedding, $s_u$, of other users to compute the negative loss term. We, however, show that under certain conditions, the model can be trained using only the positive loss term.

Let us define the positive and negative loss terms as follows:

$$\ell_{\text{pos}} = \max(0, 1 - \frac{1}{c} v_y^T W g_\theta(x)) \text{ and } \ell_{\text{neg}} = \max_{u \notin y} \frac{1}{c} v_u^T W g_\theta(x). \tag{4}$$

The positive loss term maximizes the correlation of the instance embedding with the true secret embedding, while the negative loss term minimizes the correlation with secret embeddings of other users. We have the following Lemma.

**Lemma 1.** *Assume* $\|W g_\theta(x)\| = \sqrt{c}$ *and* $v_y \in \{-1, 1\}^c$*. For* $\ell_{\text{pos}}$ *defined in (4), we have* $\ell_{\text{pos}} = 0$ *if and only if* $W g_\theta(x) = v_y$.

*Proof.* Let $z = Wg_\theta(x)$. The term $\ell_{\text{pos}} = 0$ is equivalent to $\frac{1}{c}v_y^T z \geq 1$. We have $\frac{1}{c}v_y^T z \leq \frac{1}{c}\|v_y\|\|z\| = 1$ and the equality holds if and only if $z = \alpha v_y$. Since $\|z\| = \|v_y\| = \sqrt{c}$, then $\alpha = 1$ and, hence, we have $\ell_{\text{pos}} = 0$ if and only if $z = v_y$. $\square$

The following Theorem links the positive and negative loss terms of (4) when secret vectors are chosen from ECC codewords.

**Theorem 1.** *Assume $\|Wg_\theta(x)\| = \sqrt{c}$ and $v_y \in \{-1,1\}^c$. Assume $v_i$'s are chosen from ECC codewords. For $\ell_{\text{pos}}$ and $\ell_{\text{neg}}$ defined in (4), minimizing $\ell_{\text{pos}}$ also minimizes $\ell_{\text{neg}}$.*

*Proof.* Since $v_i \in \{-1,1\}^c$, the Hamming distance between $v_{u_1}$ and $v_{u_2}$ is defined as

$$\Delta_{u_1,u_2} = \frac{1}{4}\|v_{u_1} - v_{u_2}\|^2 = \frac{1}{4}(\|v_{u_1}\|^2 + \|v_{u_2}\|^2 - 2v_{u_1}^T v_{u_2}) = \frac{c}{2}(1 - \frac{1}{c}v_{u_1}^T v_{u_2}).$$

The minimum Hamming distance between codewords is obtained as $d_{\min} = \min_{u_1 \neq u_2} \Delta_{u_1,u_2}$. As stated in Section 2.3, ECCs are designed to maximize $d_{\min}$ or, equivalently, minimize $\max_{u_1 \neq u_2} \frac{1}{c}v_{u_1}^T v_{u_2}$. Using Lemma (1), we have $\ell_{\text{pos}} = 0$ if and only if $z = v_y$, which results in $\ell_{\text{neg}} = \max_{u \notin y} \frac{1}{c}v_u^T v_y$. As a result, $\ell_{\text{neg}}$ is at its minimum when $\ell_{\text{pos}} = 0$ and $v_i$'s are chosen from ECC codewords. $\square$

Theorem (1) states that the negative loss term in (3) is redundant when $\|Wg_\theta(x)\| = \sqrt{c}$ and the secret vectors are chosen from ECC codewords, enabling the embedding-based classifiers to be trained with only the positive loss defined in (4). Note that it will still help to use $\ell_{\text{neg}}$ for training especially at early epochs, but the effect of $\ell_{\text{neg}}$ gradually vanishes as $\ell_{\text{pos}}$ becomes smaller and eventually gets close to zero. To illustrate this, we show an example of the training and test accuracy with and without $\ell_{\text{neg}}$ in Figure 4 in Appendix C. As can be seen, using $\ell_{\text{neg}}$ results in better accuracy at early epochs but does not have significant impact on the final accuracy.

## 4.2 FEDERATED USER VERIFICATION (FEDUV)

In this section, we present Federated User Verification (`FedUV`), a framework for training UV models in the federated setup. `FedUV` consists of three phases of choosing unique codewords, training, and verification, details of which are provided in the following.

**Choosing Unique Codewords.** To train the UV model with the positive loss function defined in (4), users must choose unique codewords without sharing the vectors with each other or the server. To do so, we propose to partition the space between users by the server and let users select a random message in their assigned space. Specifically, the server chooses unique binary vectors $b_u$ of length $l_b$ for each user $u \in [K]$ and sends each vector to the corresponding user. Each user $u$ then chooses a random binary vector, $r_u$, of length $l_r$, constructs the message vector $m_u = b_u\|r_u$, and computes the codeword $v_u = C(m_u)$, where $C$ is the block code.

The length of the base vectors is determined such that the total number of vectors is greater than or equal to the number of users, i.e., $l_b \geq \log_2 K$. In practice, the server can set $l_b \gg \log_2 K$ so that new users can be added to the training after training started. In experiments, we set $l_b = 32$, which is sufficient for most practical purposes. The code length is also determined by the server based on the number of users and the desired minimum distance obtained according to the estimated difficulty of the task. Using larger codewords improves the performance of the model but also increases the training complexity and communication cost of the `FedAvg` method. The proposed method has the following properties.

- It ensures that codewords are unique, because the base vectors $b_u$'s and, in turn, $m_u$'s are unique for all users. Moreover, due to the use of the ECC algorithm, the minimum distance between codewords are guaranteed to be more than a threshold determined by the code characteristics.

- The final codewords are not shared among users or with the server. Moreover, there are $2^{l_r}$ vectors for each user to choose their codeword from. Increasing $l_r$ improves the security of the method by making it harder to guess the user's codeword but reduces the minimum distance of the code for a given code length. In experiments, we set $l_r \geq 32$, which is sufficient for most practical purposes.

---

**Algorithm 2** Federated User Authentication (FedUV).    $K$: number of users, $C$: block code, $\theta, W$: model parameters, $\sigma$: a function that scales its input to have norm of $\sqrt{c}$, $q$: TPR.

---

**Codeword Selection:**
   **Server:** Send a unique binary vector, $b_u, u \in [K]$, of length $l_b \geq \log_2 K$ to user $u$
   **User** $u \in [K]$**:**
      Choose a random binary vector, $r_u$, of length $l_r$
      Construct message vector $m_u = b_u \| r_u$
      Compute codeword $v_u = C(m_u)$

**Training:**
   **Server and users:** Train UV model using FedAvg algorithm (1) and with the loss function
   $\ell_{\text{pos}} = \max(0, 1 - \frac{1}{c} v_y^T \sigma(W g_\theta(x)))$

**Warm-up Phase**$(\theta, W, v_y, q)$**:**  *// Done by users*
   Collect inputs $x'_j, j \in [n]$, and compute $e_j = \frac{1}{c} v_y^T \sigma(W g_\theta(x'_j))$
   Set $\tau$ equal to the $i$-th smallest value in $e$ where $i = \lfloor n \cdot (1 - q) \rfloor$

**Verification**$(\theta, W, v_y, \tau, x')$**:**  *// Done by users*
   $e = \frac{1}{c} v_y^T \sigma(W g_\theta(x'))$
   **if** $e \geq \tau$ **then** ACCEPT **else** REJECT

---

- The method adds only a small overhead to vanilla FL algorithms. Specifically, the server assigns and distributes unique binary vectors to users and users construct message vectors and compute the codewords.

**Training.** Figure 1 shows the model structure used in FedUV method. The model is trained using the FedAvg algorithm and with the loss function $\ell_{\text{pos}} = \max(0, 1 - \frac{1}{c} v_y^T \sigma(W g_\theta(x)))$, where $\sigma$ is a function that scales its input to have norm of $\sqrt{c}$.

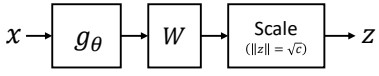

Figure 1: Model structure for FedUV.

**Verification.** After training, each user deploys the model as a binary classifier to accept or reject test examples. For an input $x'$, the verification is done as

$$\frac{1}{c} v_y^T \sigma(W g_\theta(x')) \overset{\text{accept}}{\underset{\text{reject}}{\gtrless}} \tau, \tag{5}$$

where $\tau$ is the verification threshold. The threshold is determined by each user independently such that they achieve a True Positive Rate (TPR) more than a value, say $q = 90\%$. The TPR is defined as the rate that the reference user is correctly verified. To do so, in a warm-up phase, $n$ inputs $x'_j, j \in [n]$, are collected and their corresponding scores are computed as $\frac{1}{c} v_y^T \sigma(W g_\theta(x'_j))$. The threshold is then set such that a desired fraction $q$ of inputs are verified.

Our proposed framework, FedUV, is described in Algorithm (2).

### 4.3 COMPARING COMPUTATIONAL COST OF FEDUV AND FEDAWS

FedUV has similar computational cost to FedAwS on the user side as both methods perform regular training of the model on local data (with different loss functions). On the server side, however, FedUV is more efficient, since, unlike FedAwS, it does not require the server to do any processing beyond averaging the gradients.

## 5 RELATED WORK

The problem of training UV models in federated setup has been studied in (Granqvist et al., 2020) for on-device speaker verification and in (Yu et al., 2020) as part of a general setting of FL with only positive labels. However, to the best of our knowledge, our work is the first to address the problem of training embedding-based classifiers in federated setup with only the positive loss function. Our proposed framework trains the model without sharing the secret embedding vector with the server

or other users. It is, however, not designed to defend against generic attacks in the federated setup, such as the poisoning and backdoor attacks presented in (Bhagoji et al., 2019; Bagdasaryan et al., 2020). Our method also inherits potential privacy leakage of FL methods, where users' data might be recovered from a trained model or the gradients (Melis et al., 2019). It has been suggested that adding noise to gradients or using secure aggregation methods improve the privacy of FL (McMahan et al., 2017b; Bonawitz et al., 2017). Such approaches can be applied to our framework as well.

Our approach of assigning a codeword to each user is related to distributed output representation (Sejnowski & Rosenberg, 1987), where a binary function is learned for each bit position. It follows (Hinton et al., 1986) in that functions are chosen to be meaningful and independent, so that each combination of concepts can be represented by a unique representation. Another related method is distributed output coding (Dietterich & Bakiri, 1991; 1994), which uses ECCs to improve the generalization performance of classifiers. We, however, use ECCs to enable training embedding-based classifiers with only the positive loss function.

# 6 EXPERIMENTAL RESULTS

## 6.1 DATASETS

**VoxCeleb** (Nagrani et al., 2017) is created for text-independent speaker identification in real environments. The dataset contains $1,251$ speakers' data with $45$ to $250$ number of utterances per speaker, which are generated from YouTube videos recorded in various acoustic environments. We selected $1,000$ speakers and generated $25$ training, $10$ validation and $10$ test examples for each speaker. The examples are 2-second audio clips obtained from videos recorded in one setting. We also generated a separate test set of $1,000$ examples by choosing $5$ utterances from $200$ of the remaining speakers that were not selected for training. All 2-second audio files were sampled at $8$ kHz to obtain vectors of length $2^{14}$ for model input.

**CelebA** (Liu et al., 2015) contains more than $200,000$ facial images from $10,177$ unique individuals, where each image has the annotation of $40$ binary attributes and $5$ landmark locations. We use CelebA for user verification by assigning the data of each individual to one client and training the model to recognize faces. We selected $1,000$ identities from those who had at least $30$ images, which we split into $20$, $5$ and $5$ examples for training, validation, and test sets, respectively. We also generated a separate test set with $1,000$ images from individuals that were not selected for training (one example per person). All images were resized to $64 \times 64$.

**MNIST-UV.** We created MNIST-UV dataset for user verification based on handwriting recognition. MNIST-UV examples are generated using the EMNIST-byclass dataset (Cohen et al., 2017), which contains $814,255$ images from $62$ unbalanced classes ($10$ digits and $52$ lower- and upper-case letters) written by $3,596$ writers. A version of this dataset, called FEMNIST, has been used to train a 62-class classifier in federated setup by assigning the data of each writer to one client (Caldas et al., 2018). In FEMNIST, the difference in handwritings is used to simulate the non-iid nature of the clients' data in federated setup.

We repurpose EMNIST for the task of user verification by training a classifier that recognizes the handwritings., i.e., similar to FEMNIST, the data of each writer is assigned to one client but the model is trained to predict the *writer IDs*. To this end, we created MNIST-UV dataset that contains data of $1,000$ writers each with $50$ training, $15$ validation, and $15$ test examples. Each example in the dataset is of size $28 \times 28 \times 4$ and is composed of images of digits $2, 3, 4$ and $5$ obtained from one writer. For each writer, the training examples are unique; however, the same sub-image (images of digits $2, 3, 4$ or $5$) might appear in several examples. This also holds for validation and test sets. The sub images are, however, not shared between training, validation, and test sets. We also generated a separate test set with $1,000$ examples from writers that were not selected for training (one example per writer). Figure 3 in Appendix A shows few examples of the MNIST-UV dataset.

## 6.2 EXPERIMENT SETTINGS

**Training setup.** We train UV models using the `FedAvg` method with $1$ local epoch and $20,000$ rounds with $0.01$ of users selected at each round. The model structures are provided in Appendix B.

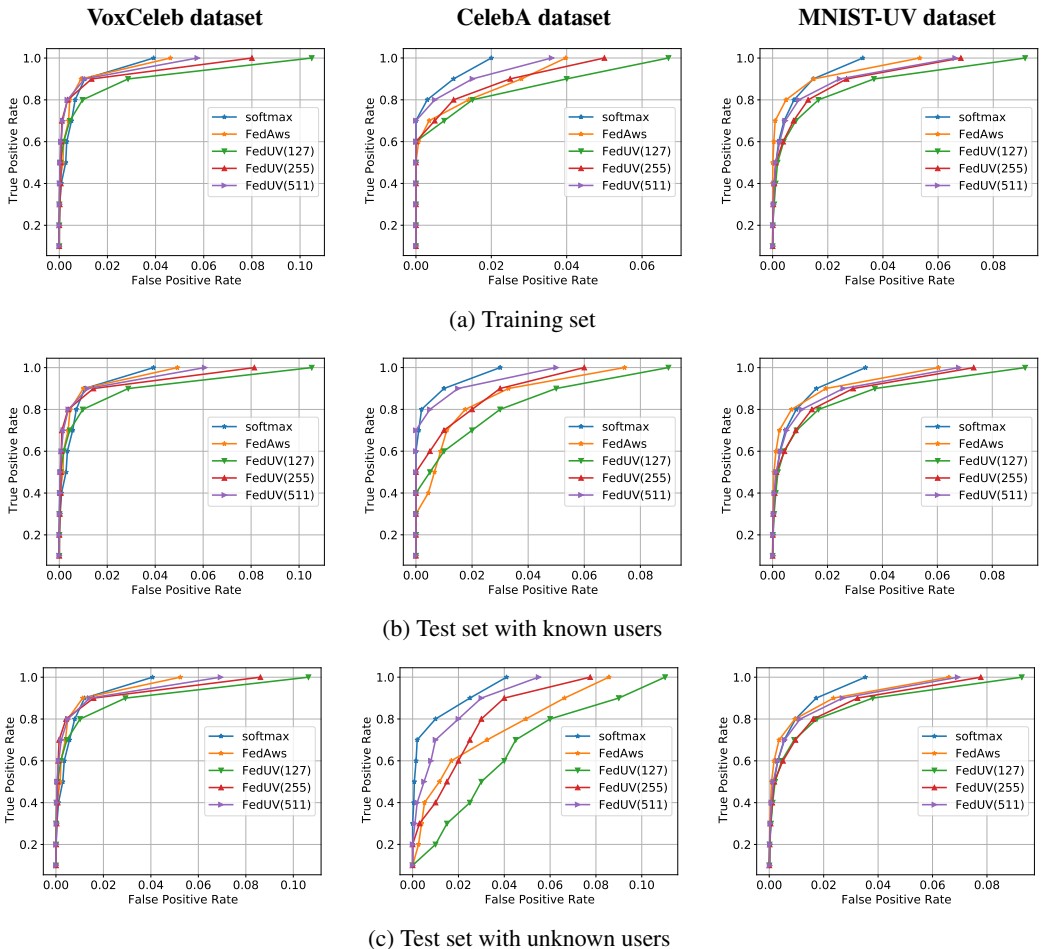

Figure 2: ROC curves for models trained in federated setup using `softmax`, `FedAwS` and `FedUV` algorithms. `FedUV` ($c$) denotes `FedUV` with code length of $c$. It can be seen that `FedUV` performs on par with `FedAwS`, while `softmax` outperforming both methods. Also, as expected, increasing the code length improves the performance of `FedUV` algorithm. Note that, unlike `FedUV`, both `softmax` and `FedAwS` share embeddings with other users and/or the server.

**Baselines.** We compare our `FedUV` method with the `FedAwS` algorithm (Yu et al., 2020) and regular federated learning with the softmax cross-entropy loss function, which we refer to as `softmax` algorithm. Note that `softmax` and `FedAwS` share the embedding of each user with other users and/or the server. Similar to `FedUV`, we perform a warm-up phase for the two baselines to determine the verification threshold for each user.

**Generating codewords.** We use BCH coding algorithm to generate codewords of lengths $127, 255$ and $511$. The code lengths are chosen to be smaller than the number of users $(1,000)$ to emulate the setting with a very large number of users. For each code length, we find the message length of greater than or equal to $64$ that produces a valid code. Table 1 shows the code statistics.

Table 1: Statistics of BCH codewords used in experiments.

| Code length | Message length | $d_{\min}$ |
|---|---|---|
| 127 | 64 | 21 |
| 255 | 71 | 59 |
| 511 | 67 | 175 |

## 6.3 VERIFICATION RESULTS

We evaluate the verification performance on three datasets, namely 1) training data, 2) test data of users who participated in training, and 3) data of users who did not participate in training. Figure 2 shows the ROC curves. The verification performance is best on training data and slightly degrades when the model is evaluated on test data of users who participated in training and further reduces

on data of new users. All methods, however, achieve notably high TPR, e.g., greater than $80\%$, at low False Positive Rates (FPRs) of smaller than $10\%$, implying that the trained UV models can reliably reject the impostors. The regular `softmax` training outperforms both `FedAwS` and `FedUV` algorithms in most cases, especially at high TPRs of greater than $90\%$. `FedUV`'s performance is on par with `FedAwS`, while not sharing the embedding vectors with the server. Also, as expected, increasing the code length in `FedUV` improves the performance.

## 7 CONCLUSION

In this paper, we presented `FedUV`, a framework for private and secure training of user verification models. In `FedUV`, users first collaboratively choose unique secret vectors from codewords of an error-correcting code and then train the model using `FedAvg` method with a loss function that only uses their own vector. After training, each user independently performs a warm-up phase to obtain their verification threshold. We show our framework addresses the problem of existing approaches where embedding vectors are shared with other users or the server. Our experimental results for user verification with voice, face, and handwriting data show `FedUV` performs on par with existing approaches, while not sharing the embeddings with other users or the server.

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

## A  MNIST-UV DATASET

Figure 3 shows examples from MNIST-UV dataset. Note that, in figure, sub-images are placed in a $2 \times 2$ grid for clarity.

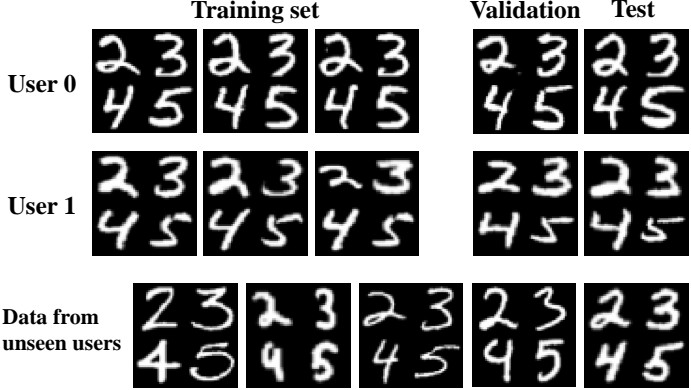

Figure 3: Examples from MNIST-UV dataset created for user verification by handwriting. Each example in the dataset is of size $28 \times 28 \times 4$ and is composed of images of digits $2, 3, 4$ and $5$ obtained from one writer. In figure, sub-images are placed in a $2 \times 2$ grid for clarity. MNIST-UV dataset contains data of $1,000$ writers each with 50 training, 15 validation, and 15 test examples. It also contains a separate test set with $1,000$ examples from writers that were not selected for training (one example per writer).

## B  EXPERIMENTAL SETUP

Table 2 shows network architectures used for each dataset. In models, we use Group Normalization (GN) instead of batch-normalization (BN) following the observations that BN does not work well in non-iid data setting of federated learning (Hsieh et al., 2019). The models are trained with SGD optimizer with learning rate of $0.1$ and learning rate decay of $0.01$.

Table 2: Network architectures for training UV models with different datasets. convxd$(c1, c2, k, p)$ is x-dimensional convolutional layer with $c1$ and $c2$ input and output channels, respectively, kernel size of $k$ and padding of $p$. The default value of $p$ is 1. GN$(G)$ is group normalization layer with $G$ groups. `Scaling` layer scales its input to have norm of $\sqrt{c}$. $c$ is the code length in case of `FedUV` and the number of users in `softmax` and `FedAwS` algorithms.

| VoxCeleb | CelebA | MNIST-UV |
|---|---|---|
| conv1d$(1, 64, k = 15)$ | conv2d$(3, 64, k = 3)$ | conv2d$(4, 64, k = 3, p = 3)$ |
| relu, max_pool1d$(4)$, GN$(2)$ | relu, max_pool2d$(2)$, GN$(2)$ | relu, max_pool2d$(2)$, GN$(2)$ |
| conv1d$(64, 128, k = 9)$ | conv2d$(64, 128, k = 3)$ | conv2d$(64, 128, k = 3, p = 1)$ |
| relu, max_pool1d$(8)$, GN$(2)$ | relu, max_pool2d$(2)$, GN$(2)$ | relu, max_pool2d$(2)$, GN$(2)$ |
| conv1d$(128, 256, k = 7)$ | conv2d$(128, 256, k = 3)$ | conv2d$(128, 256, k = 3, p = 1)$ |
| relu, max_pool1d$(8)$, GN$(2)$ | relu, max_pool2d$(2)$, GN$(2)$ | relu, max_pool2d$(2)$, GN$(2)$ |
| conv1d$(256, 512, k = 5)$ | conv2d$(256, 512, k = 3)$ | conv2d$(256, 512, k = 3, p = 1)$ |
| relu, max_pool1d$(8)$, GN$(2)$ | relu, max_pool2d$(2)$, GN$(2)$ | relu, max_pool2d$(2)$, GN$(2)$ |
| conv1d$(512, 1024, k = 3)$ | conv2d$(512, 1024, k = 3)$ | conv2d$(512, 1024, k = 3, p = 1)$ |
| relu, max_pool1d$(8)$, GN$(2)$ | relu, max_pool2d$(4)$, GN$(2)$ | relu, max_pool2d$(2)$, GN$(2)$ |
| Flatten | Flatten | Flatten |
| FC$(1024, c)$ | FC$(1024, c)$ | FC$(1024, c)$ |
| `Scaling` // for `FedUV` | `Scaling` // for `FedUV` | `Scaling` // for `FedUV` |

## C   TRAINING WITH AND WITHOUT $\ell_{\mathrm{neg}}$

Figure 4 shows the training and test accuracy with and without $\ell_{\mathrm{neg}}$ for MNIST-UV dataset (For the sake of simplicity, here we show the accuracy rather than the TPR and FPR). As seen, using $\ell_{\mathrm{neg}}$ results in better accuracy at early epochs but does not have significant impact on the final accuracy.

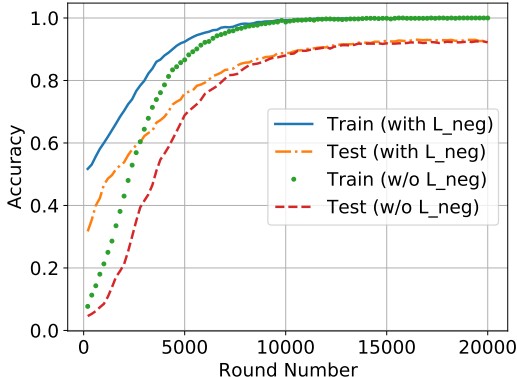

Figure 4: Training and test accuracy with and without $\ell_{\mathrm{neg}}$ for MNIST-UV dataset.

## D   AUC

Table 3 shows the Area Under the Curve (AUC) of the ROC curves of Figure 2. As seen, the AUC is above 0.99 in most cases, indicating that all methods perform well on different datasets.

Table 3: AUC of the ROC curves of Figure 2.

| Dataset | Dataset type | softmax | FedAwS | FedUV (127) | FedUV (255) | FedUV (511) |
|---------|--------------|---------|--------|-------------|-------------|-------------|
| VoxCeleb | train | 0.997 | 0.998 | 0.995 | 0.998 | 0.998 |
| | test (known) | 0.997 | 0.998 | 0.995 | 0.998 | 0.998 |
| | test (unknown) | 0.997 | 0.998 | 0.995 | 0.998 | 0.998 |
| CelebA | train | 0.999 | 0.995 | 0.994 | 0.996 | 0.998 |
| | test (known) | 0.999 | 0.992 | 0.989 | 0.994 | 0.998 |
| | test (unknown) | 0.996 | 0.981 | 0.969 | 0.986 | 0.992 |
| MNIST-UV | train | 0.997 | 0.998 | 0.993 | 0.995 | 0.996 |
| | test (known) | 0.996 | 0.997 | 0.993 | 0.994 | 0.995 |
| | test (unknown) | 0.996 | 0.996 | 0.992 | 0.993 | 0.995 |

