# OpenReview forum: "Secure Federated Learning of User Verification Models"
_ICLR.cc/2021/Conference — Reject_

### Official Review · AnonReviewer3 · 2020-10-14
**Secure Federated Learning of User Verification Models**

**Rating:** 7
**Confidence:** 5

**Review:**

In this paper, the authors focus on designing a federated user verification solution. Specifically, the authors address two fundamental challenges associated with user verification, i.e., one-class data (positive data only), and privacy protection (i.e., the raw data and the embeddings of the users and class). Technically, the authors extend a very recent work called FedAWS by (Yu et al., 2020), and introduce a user-specific codewords, which not only protect users' privacy (i.e., not sharing the embedding with other users or the server) but also do not need the negative samples (i.e., the two loss functions in Eq.(5) reduces to one due to equivalence shown in Theorem 1). We can see that the main idea of re-writing the Eq.(2) into two loss functions in Eq.(4) and Eq.(5), and introducing codewords are novel and effective, which also address the two challenges well.

Empirical studies on three user verification cases show the effectiveness of the proposed solution FedUV.

Overall, the technique is novel (and I like this idea) and the paper is well presented. I recommend acceptance.

---

### Official Review · AnonReviewer2 · 2020-10-19
**Nearly flawless paper**

**Rating:** 6
**Confidence:** 4

**Review:**

**Summary**
Federated learning takes advantage of the fact that private user data does not need to be transferred and shared across devices or servers. This makes FL particularly attractive for the user verification scenario, where privacy-sensitive biometric data are used to train verification models. One crucial hurdle in this scenario is that per device, only positive data are present, potentially turning the device-wise training objective ill-posed (all embedding are likely to collapse to a single point). As a way to introduce negative examples, FEDAWS has been developed and presented at ICML 2020. This paper recognizes a crucial security risk in the FEDAWS system, that embeddings of user data are transferred to the server, and proposes a more secure training methodology, FEDUV, that involves the error-correcting codes. FEDUV enjoys stronger security guarantees while showing comparable ROC curves as FEDAWS at nearly identical computational costs (though not entirely sure about the computational cost bit ;) ).

**Pros**
The motivation is spot on. Having to see any form of negative samples is the itchy point of the FL-based user verification system. FEDUV magically solves this issue by pre-defining a unique prototype vector for each user, which are not shared across users and are by design far apart from each other (this is the crucial trick!) by employing a technique in error-correcting codes (ECC). As a result, each user's endeavour to get closer to the own prototype vector ensures the maximisation of distance from the others' prototype vectors.

Three experiments that are quite close to real-world scenarios (speaker, face, and handwriting-based verification) show that the performance of FEDUV is comparable to FEDAWS, the state of the art framework from ICML 2020 with weaker security guarantees.

Writing is nearly flawless. Highly enjoyable paper.

**Cons**
No major cons. Perhaps explain in a bit more depth on the BCH code to illustrate (at least a high-level, hand-wavy description) how it assigns the codes in a distance-maximizing manner. Section 2.3 only explains the desiderata for BCH, rather than *how* BCH achieves it. Please also confirm that FEDUV spends nearly identical computational cost as FEDAWS. Somehow I got this from the paper, but have not found a solid reference that confirms this (if not, please explain, too).

Nits: Please add grid lines and row titles (training set, test set with known users, test set with unknown users) in Figure 2 plots. Baslines --> Baselines. Flatten the last part of Section 1 as paragraphs rather than itemize? Yu et al. 2020 (FEDAWS) is an ICML paper, not arXiv - please fix the reference.

**Key reasons for the rating**
I don't find any major rationale to reject this paper. However, its novelty is also eclipsed by the Yu et al. 2020 (FEDAWS) paper. Though I really like this paper, I believe the best scores should be reserved for more innovative papers.

**After rebuttal & discussion**
I still tend to think that the paper's scope can be adjusted relatively easily (it is not too difficult to insert more disclaimers and change the title), and we can force apply the adjustment by conferring a conditional acceptance.

But I'm sold on the point that there is a lack of argumentation on whether undisclosing the user-specific embedding will improve the privacy guarantee. I had taken this argument as granted, but this is indeed not so obvious, given that there exist many attacks that are applicable in this kind of scenario, as R4 has argued. It would be great if the authors could quantify the improved privacy guarantee.

I'm okay with rejecting the paper then. I still like the paper quite a lot, but rejecting it will also give the authors a good chance to assimilate more points of views in the paper.

---

> ### Author Response · Authors · 2020-11-19
> **Response to Reviewer 2**
>
> Thank you for your comments. Our responses are provided in the following.
>
> * On BCH code. We will add a description of how the BCH codewords are constructed to the paper. Please, however, note that the choice of the coding algorithm is not crucial to our work and our method works with any error-correcting coding algorithm. We used BCH code because it provides codes with a wide range of message and code lengths.
>
> * On computational cost. FedUV has similar computational cost as FedAWS on the client-side. On the server-side, however, FedUV is more efficient, since, unlike FedAWS, it does not require the server to do any processing beyond averaging the gradients. We will add a more detailed discussion on the computational cost comparisons to the experimental results section.
>
> * Thanks for your helpful edit suggestions. We will revise the paper accordingly.

---

### Official Review · AnonReviewer4 · 2020-10-27
**unclear security and privacy guarantees**

**Rating:** 2
**Confidence:** 4

**Review:**

The paper leverages federated learning to train user verification models. The authors claim that their new federated learning addresses the security and privacy issues of previous methods. In particular, for privacy, the users do not need to send their class embedding vectors to server nor other users. For security, the paper claims that the proposed method is secure against poisoning attacks and evasion attacks.

Strengths

I think the major strength of the paper is to design a loss function and a way of modeling embedding vectors for users such that the embedding models can be learnt without sharing the embedded vectors to the server nor other users.

Weaknesses

The paper is weak on its security and privacy claims.

1. For privacy, can you quantify the privacy leakage of sharing embedded vectors with the server? Without a formal quantification, it is hard to claim your method is more private.

2. Poisoning attack. I don't think the paper addresses the poisoning attacks. The paper considers that the server may poison the learnt model. However, in the proposed method, the server can still poison the model. In particular, the server can send arbitrary new model to each user. In general, it is hard to defend against malicious server who performs poison attacks.

Also, malicious users can poison the model training, which are more realistic poisoning attacks. But such poisoning attacks are not considered. I don't see how the proposed method can address these poisoning attacks.  Some references on poisoning attacks:

https://arxiv.org/abs/1807.00459

https://arxiv.org/abs/1911.11815

https://openreview.net/forum?id=rkgyS0VFvr

3. Evasion attack. The proposed cannot address evasion attack at all.

4. Experimetal details. Can you add more details on experimental details, e.g., learning rate. How is experiment on softmax loss function implemented.

5. Can you also report AUC to compare different methods, since you already show the true positive rate vs. false positive rate curves?

---

> ### Author Response · Authors · 2020-11-19
> **Response to Reviewer 4**
>
> Thank you for your comments. Our responses are provided in the following.
>
> * On privacy quantification. Our paper focuses on the security problems associated with the leakage of the embedding vector. Please note that, unlike the input biometric data, the embedding vector is not private per se. It is, however, highly security-sensitive since it will be used for user verification. In our paper, we proposed a method to train the model without sharing the embedding vector with the server (or other users). As a result, our method provides robustness against the attacks that require access to the embedding vector.
>
> * On security claims. Please note that the goal of our paper is not designing a model robust against generic evasion and poisoning attacks, and, indeed, as you mentioned, a model trained with FedUV will certainly be vulnerable against adversarial examples and poisoning attacks like any other deep neural network. Our goal, instead, is to prevent the attacks that specifically result from the leakage of the embedding vector. We provided examples of how sharing the embedding with the server allows the server to run special types of poisoning and evasion attacks, and showed that our method provides robustness against such attacks.
>
> * About poisoning attack:
>   * Response to “the server can send arbitrary new model to each user”: such an attack can be detected by users since they will notice that the accuracy of new global models on their validation data is not improving as the training progresses.
>   * Response to poisoning attack by malicious users: in poisoning attacks, the attacker usually aims to manipulate the training data such that the model misclassifies specific test examples or classifies them into a target class. In our problem of user verification, each user is associated with one class. Hence, for the server or users to run poisoning attack, they need to force the model to either (i) reject the correct examples of a target user, or (ii) verify some fake examples as true inputs of the target user. Attacks (i) and (ii) need access to input examples and the embedding vector of the target user, respectively. The use of federated learning eliminates the need to share input examples and our method, FedUV, eliminates the need to share embedding vectors. Hence, our method is robust against such poisoning attacks.
>
> * About evasion attack. In the context of user verification models, evasion attack can be used to force the model to verify fake examples. The attack can be described as follows: manipulate a fake example such that the model will verify it as a true example generated by a target user. To do so, the attacker needs to modify the fake example such that the model will output the embedding vector of the target user. The embedding vectors, however, are not shared with the server of other users. Hence, such an attack is not feasible.
>
> * On experimental details:
>   * We train UV models using the FedAvg method with one local epoch and $20,000$ rounds with $0.01$ of users selected at each round. The models are trained with SGD optimizer with learning rate of $0.1$ and learning rate decay of $0.01$. We will provide more details on the experimental setup in the paper.
>   * The softmax loss function corresponds to the regular training of a multi-class classifier, i.e., each user is assigned a class ID and they collaboratively train the entire model (including the last FC layer as embedding vectors).
>
> * On reporting AUCs. Thanks for your suggestion. We will add the AUC of different methods in the revised version of the paper.

---

> > ### Comment · AnonReviewer4 · 2020-11-19
> > **Security/privacy concerns are not addressed**
> >
> > Privacy quantification -- the paper claims the proposed method is more private, which is contradictory to the response which claims that the paper focuses on security. Tune down your claim in the paper.
> >
> >  "Please note that the goal of our paper is not designing a model robust against generic evasion and poisoning attacks". Again, this is contradictory to the claims in the paper. You explicitly mentioned evasion and poisoning attacks. If these are not your goals, the paper should be revised.
> >
> > Response to “the server can send arbitrary new model to each user”: such an attack can be detected by users since they will notice that the accuracy of new global models on their validation data is not improving as the training progresses. -- this is not correct. A backdoor attack does not influence normal model accuracy. An "arbitrary new model" means that the server can carefully design a model and send it to each user.
> >
> >
> > Response to poisoning attack by malicious users: in poisoning attacks, the attacker usually aims to manipulate the training data such that the model misclassifies specific test examples or classifies them into a target class. In our problem of user verification, each user is associated with one class. Hence, for the server or users to run poisoning attack, they need to force the model to either (i) reject the correct examples of a target user, or (ii) verify some fake examples as true inputs of the target user. Attacks (i) and (ii) need access to input examples and the embedding vector of the target user, respectively. The use of federated learning eliminates the need to share input examples and our method, FedUV, eliminates the need to share embedding vectors. Hence, our method is robust against such poisoning attacks.
> >
> > -- "in poisoning attacks, the attacker usually aims to manipulate the training data" is also not correct for federated learning. An attacker does not need to manipulate training data in federated learning. A malicious user can directly send carefully designed model updates to the server to attack the learnt model. The malicious users do not need input examples of the target user.
> >
> >
> > About evasion attack. In the context of user verification models, evasion attack can be used to force the model to verify fake examples. The attack can be described as follows: manipulate a fake example such that the model will verify it as a true example generated by a target user. To do so, the attacker needs to modify the fake example such that the model will output the embedding vector of the target user. The embedding vectors, however, are not shared with the server of other users. Hence, such an attack is not feasible.
> >
> > -- This discussion is also incorrect to me. An attacker does not need to know the embedding vectors to perform evasion attacks. An attacker can just perform black-box evasion attacks. Starting from a fake example and gradually adjust it until the model classifies it as the target user.
> >
> >  I still think this paper's claims on security/privacy are unjustified. The technique itself is interesting to learn user verification models, but it is far from "secure federated learning" of user verification models. It is an interesting "federated learning" but not "secure federated learning" of user verification models. I would suggest the authors to change the title to be "federated learning of user verification models", tune down the claims on security and privacy, and acknowledge the security limitations.

---

> > > ### Author Response · Authors · 2020-11-19
> > > **Response to Security/Privacy Concerns**
> > >
> > > Thank you for your comments. Our responses are provided in the following.
> > >
> > > * About privacy and security focus of the work. We outlined the privacy and security requirements of the user verification (UV) applications in the first two paragraphs of the Section 3-1. It is mentioned that the input biometric data are privacy-sensitive, and the embedding vectors are security-sensitive. We used federated learning to address the privacy problem of sharing the input data and proposed a method based on error-correcting codes to tackle the security problem of sharing embedding vectors.
> > >
> > > * About poisoning and evasion attacks. In the third paragraph of the Section 3-1, we explained the security threats of the leakage of the embedding vector. We also provided examples of the relevant poisoning and evasion attacks, where the attacker uses the knowledge of the embedding vector to run the attack. The goal of the paper is to train the UV model without sharing the embedding vector, so that the model will be robust against such attacks that need access to the embedding vector.
> > >
> > > * About backdoor attach by the server. Can you please explain how a backdoor attack can be performed without the knowledge of the embedding vectors of the users?
> > >
> > > * About poisoning attack by malicious users. Can you please explain how malicious users can perform a poisoning attack without the knowledge of the embedding vectors of other users?
> > >
> > > * About evasion attack. You mentioned that the attacker can start “from a fake example and gradually adjust it until the model classifies it as the target user.” Can you please explain how the attack can be performed without the knowledge of the embedding vector of the target user?

---

> > > > ### Comment · AnonReviewer4 · 2020-11-20
> > > > **Lack understanding of security issues**
> > > >
> > > > Thanks for the response.
> > > >
> > > > For backdoor attacks, you can refer to the references I mentioned in my review (may not be exactly for your setting, but the attack is general).
> > > >
> > > > For evasion attack, you can refer to the following paper:
> > > >
> > > >  Decision-Based Adversarial Attacks: Reliable Attacks Against Black-Box Machine Learning Models. https://arxiv.org/abs/1712.04248. ICLR, 2018.
> > > >
> > > > For any model (e.g., user verification model), an attacker can perform evasion attack via just querying the model. It does not matter whether the attacker has access to the embedding vector or not.
> > > >
> > > > Based on the authors' response, I feel the authors misunderstood evasion attacks and poisoning attacks, although they claimed that their method is secure against them. Moreover, it seems like that the authors are not willing to revise their paper to address my comments on these issues. Therefore, I downgraded my rating score.

---

> > > > > ### Author Response · Authors · 2020-11-20
> > > > > **Response to Security Issues**
> > > > >
> > > > > Thank you for your comments. Our responses are provided in the following.
> > > > >
> > > > > * About attacks. Please note that our method is designed to provide robustness against the attacks on user verification (UV) models. Studying generic attacks on federated learning is out of scope of the paper. We will make this clearer in the revision.
> > > > >
> > > > > * About backdoor attack by the server. Generic backdoor attacks can be, indeed, performed, e.g., to force the model to map a fake example $x$ into a fake embedding vector $y$. Such an attack, however, will not negatively affect the verification performance of the model. The reason is that the server does not have access to the input data or the embedding vectors of users and cannot design the fake input or the fake embedding vector in a way to harm a target user. We will make this clearer in the revision.
> > > > >
> > > > > * About evasion attack.
> > > > >   * As we mentioned in previous responses, in the context of user verification models, evasion attacks can be used to force the model to verify fake examples. You also mentioned that the attacker can start “from a fake example and gradually adjust it until the model classifies it as the target user.”
> > > > >   * In our proposed method, the verification is done according to Equ. (6), by comparing the output of the model with a secret binary vector $v$. That is, the model returns a binary value (accept or reject) for each query. In experiments, we used codewords of length $127$, $255$, and $511$ as vector $v$.
> > > > >   * Since the vector $v$ is not known to the attacker, gradient-based attacks cannot be performed. As you mentioned, an alternative approach for the attacker is to just query the model. However, the probability of forcing the model to verify a perturbed input is very small, as there are $2^v$ number of possible vectors. Please note that UV models are usually deployed on edge devices for applications such as unlocking the phone and it is not possible to make large number of queries to them in real-world settings. We will include this discussion in the revised version of the paper to clarify our threat model.

---

### Official Review · AnonReviewer1 · 2020-10-29
**Secure federated learning user verification model training**

**Rating:** 8
**Confidence:** 3

**Review:**

The authors propose a method that allows training of UV methods without sharing any user (exemplar or class) embeddings with the server or other uses. Models are trained using gradient averaging on the server, so any leakage through that is not addressed in this work. The paper shows experimental results on speaker identification, face and handwriting verification tasks. The authors argue that this is the first work that considers secure training in a federated setup, with neither raw inputs nor exemplar or class embeddings being shared with the server or other users.

#### Pros

* The paper is clearly written and the derivations are sound (for the most part, see questions below).
* The idea appears to be novel and a significant delta compared to the SoTa in terms of security and the novelty of a secure embedding learning protocol in the federated setup were only (one) positive classes are available for training.
* The experimental results are promising albeit can't compete with existing less secure methods.

#### Cons

- Clarity of experiments
  - Especially for the face verification task the code length seems to play a major role. Any discussion giving an understanding of this would be appreciated. Specifically, how and why does $d_{min}$ affect the accuracy. Bottom of page 5 mentions that increasing the code-words and presumably $d_{min}$ increases the performance, but no reasoning is provided.
  - Additional insights of how the baselines (softmax, FedAws) were trained and what the emedding sizes are would be helpful. Is the embedding size ~64 in all cases?

#### Questions & Comments

- The assumption of $||z|| = \sqrt{c}$ should be put into context. What are the practical applications for this assumption. Is it merely there for the math to work out?
- The theorems show that $l_{neg}$ is redundant for when $l_{pos}=0$, however, it is not clear to me that minimizing $l_{pos}$ also corresponds to minimizing $l_{neg}$. In practice, $l_{pos}$ will likely never reach $0$ and a negative loss term could have a significant contribution to the loss surface.
- Page 6 mentions that increasing $l_r$ reduces the minimum distance of the code for a given code length. Why is this the case? Is it because $r_u$ is sampled by the clients and no guarantees can be made? A more detailed discussion would be helpful.

This work proposes a new idea that allows training embeddings for verification with only positive classes in a federated setting, while ensuring security. Some areas could be clarified in the paper, especially why it is sufficient to proof the redundancy of the negative loss term only for the global minimum of when $l_{pos}=0$. Assuming the authors can provide a satisfying explanation, I recommend accepting this work.

---

> ### Author Response · Authors · 2020-11-19
> **Response to Reviewer 1**
>
> Thank you for your comments. Our responses are provided in the following.
>
> * About $d_{min}$:
>      * Larger code length results in larger $d_{min}$, which is the minimum difference between any two codewords. In communication systems, the difference of $d_{min}$ between codewords can be used to correct up to $\lfloor(d_{min}-1)/2\rfloor$ errors (for binary codewords). For example, for $d_{min}=25$, if a received codeword contains up to $12$ errors, it can be correctly decoded since the true codeword is the closest one to it.
>      * In our application, we used codewords as output representations of the model. Hence, the model can be viewed as a communication channel that makes errors at test time. If the minimum distance between the codewords is larger, the model can make more errors and still correctly classify the input, which in turn results in higher test accuracy.
>
> * About baselines. The model architectures used for each dataset are provided in Appendix A. The embedding size of softmax and FedAWS is $1024$ in all cases. The softmax baseline is the regular training using the FedAvg method and without any security constraint. In FedAWS, users send their embedding vectors to the server and the server maximizes the pairwise distances between embeddings in addition to gradient averaging.
>
> * About assumption of $\|\|z\|\|=\sqrt{c}$. The network is trained to maximize the correlation of the instance embedding (model output) with the codeword. To make the optimization easier, the model output is scaled to have the same norm as the codeword. The normalization is important for the proofs but is also crucial in experiments as we observed that it increases the convergence rate and also improves the final accuracy.
>
> * About $l_{pos}$ and $l_{neg}$. It will indeed help to use $l_{neg}$ for training especially at early training rounds, but the effect of $l_{neg}$ gradually vanishes as $l_{pos}$ becomes smaller and eventually gets close to zero. To illustrate this, we show the training and test accuracy with and without $l_{neg}$ for MNIST-UV dataset (Please note that for the sake of simplicity, here we show the accuracy rather than the TPR and FPR). The figure can be found [[here]](https://drive.google.com/file/d/1BosJLUymhVNJqFli70iQkk3Wdi8WeRFT/view?usp=sharing). As can be seen, using $l_{neg}$ results in better accuracy at early epochs but does not have significant impact on the final accuracy.
>
> * On the relationship of $l_r$ and the minimum distance: The codewords are constructed as $v=C(m)$, where $C$ is the coding algorithm and $m=b\|\|r$ is the message vector, in which $b$ is a unique binary vector assigned by server to each user and $r$ is a random binary vector chosen by each user. Let $c$, $l_m$, $l_b$ and $l_r$ be the lengths of the codeword, message vector, unique ID vector and random vector, respectively. There are $2^{l_m}$ distinct codewords. With larger $l_m$ and the same $c$, there will be more codewords in the same space size and so their pairwise distance decreases. Increasing $l_r$ increases $l_m$ and, hence, reduces the minimum distance of the code. We will add more details about this to the paper.

---

### Author Response · Authors · 2020-11-23
**Paper is revised**

We would like to thank the reviewers for their valuable feedback. We have updated the paper to address the comments.

The major changes made to the paper are as follows:
* We explicitly stated that the goal of the paper is to train embedding-based classifiers in federated setup with only the positive loss term. We removed any statement that implied our method addresses the problem of poisoning or evasion attacks in FL and added that "Our proposed framework trains the model without sharing the secret embedding vector with the server or other users. It is, however, not designed to defend against generic attacks in the federated setup, such as the poisoning and backdoor attacks presented in (Bhagoji et al., 2019; Bagdasaryan et al., 2020)."
* We added a figure to Appendix showing the accuracy with and without $l_{neg}$ and explained that including $l_{neg}$ in training helps with accuracy at early epochs but does not have significant impact on the final accuracy.
* We added AUC of the ROC curves of Figure 2 to the appendix.

---

### Decision · Program_Chairs · 2021-01-07
**Final Decision**

**Decision:**

Reject

**Comment:**

In this paper, the authors propose to adapt the recent paper by Yu et al. (ICML 2020), namely FedAwS. In that paper, the authors solved a potential failure mode in federated learning, when all the users only have access to one class in their devices. In this paper, the authors extend FedAwS to a setting in which federated learning is used for User Verification (UV), namely FedUV. The authors argue that the previous paper could not be the solution to learning UV because FedAwS share the embedding vectors with the server.

The authors then show a procedure in which they can learn a classifier in which the embedding vectors are not needed to be shared with the classifier. They use error-correcting codes to make the mapping sufficiently different and that allows the training to succeed without sharing the embedding. The proposed change is only marginally worse than FedAwS and centralized learning. This is the part of the paper that has attracted positive comments and is praised by all the reviewers.

The authors take as given that by not sharing the embedding vectors and by using randomly generated error-correcting codes, the whole procedure is privacy-preserving and secure. The 4th reviewer indicates that these guarantees need to be proven and points out several references that hint toward flaws in the argument by the authors. Reviewer 4th does say that not sharing the embeddings might not be enough, but that self-evident arguments are not enough.

This paper provides a significant improvement for a federated machine learning algorithm that deserves publication, but the rationale of the paper is flawed from a privacy and security viewpoint. I think if the paper is published as is, especially with the proposed title, it will create a negative reaction by the security and privacy community for not adhering to their standards. We cannot lower those standards.

I suggest to the authors that they can follow two potential paths for publishing this work:

1 Change the scope of their algorithm. For example, I can imagine that by not sharing the embedding the communication load with the server might be significantly reduced or that adding new users with new classes can be easier.

2 Follow the recommendation from Reviewer 4 and show that the proposed method is robust against the different attacks.

Minor comments:

For a paper that is trying to solve the AU problem, I would expect a discussion about why learning is better than a private algorithm. In a way, learning is sharing, and that increases the risk of mischief by malicious users.

The discussion about error-correcting codes and the minimum distance is quite old fashion. In high dimensions, the minimum distance is not the whole story. LDPC codes make sense when we stop focusing on minimum distance codes and minimum distance decoding. I would recommend having a look at the Berlekamp’s Bat discussion in David MacKay’s book (Chapter 13).